# Pigment analysis based on a line-scanning fluorescence hyperspectral imaging microscope combined with multivariate curve resolution

Lijin Lian[1,2], Xuejuan Hu[1,2,3]*, Zhenhong Huang[1,2,3], Liang Hu[1,2,3], Lu Xu[2,3]

**1** Key Laboratory of Advanced Optical Precision Manufacturing Technology of Guangdong Provincial Higher Education Institute, Shenzhen Technology University, Shenzhen, Guangdong, China, **2** Guangdong Provincial Key Laboratory of Micro/Nano Opticmechatronicas Engineering, Shenzhen University, Shenzhen, Guangdong, China, **3** Sino-German College of Intelligent Manufacturing, Shenzhen Technology University, Shenzhen, Guangdong, China

* huxuejuan@sztu.edu.cn

**Data Availability Statement:** All relevant data are within the paper and its Supporting information files.

**Funding:** The research was funded by National Natural Science Foundation of Guangdong, China

## Abstract

A rapid and cost-effective system is vital for the detection of harmful algae that causes environmental problems in terms of water quality. The approach for algae detection was to capture images based on hyperspectral fluorescence imaging microscope by detecting specific fluorescence signatures. With the high degree of overlapping spectra of algae, the distribution of pigment in the region of interest was unknown according to a previous report. We propose an optimization method of multivariate curve resolution (MCR) to improve the performance of pigment analysis. The reconstruction image described location and concentration of the microalgae pigments. This result indicated the cyanobacterial pigment distribution and mapped the relative pigment content. In conclusion, with the advantage of acquiring two-dimensional images across a range of spectra, HSI conjoining spectral features with spatial information efficiently estimated specific features of harmful microalgae in MCR models.

## Introduction

The harmful algae boom is an environmental problem which affects water quality and alters parameters such as oxygen qualification and light penetration that are crucial to other biologies [1]. Rapid and reliable detection systems for algae are becoming important. Conventional methods of algae detection and analysis are time-consuming and require specialized equipment. One pigment analysis approach is to detect the fluorescent signal using UV-visible spectrophotometer [2]. The method has a preparation experiment that extracting pigment from algal. Another approach is to capture images based on specific fluorescence signatures using specialized microscope [3]. That is means that the value of fluorescent peaks is considered as the gold standard for pigment analysis.

(2017A030310568), Science and Technology Program of Shenzhen, China (JCYJ20180301170959233) and Special Project for Science and Technology Innovation Funds of Pingshan District, Shenzhen(PSKG202006). But the funders had no role in study design, data collection and analysis, decision to publish, or preparation of the manuscript.

**Competing interests:** The authors have declared that no competing interests exist.

Previous studies have found that pigments extracted from microalgae mainly include chlorophyll, carotenoids, astaxanthin and phycocyanin [4]. The main problem of pigment extraction is irreversible damage to the algae. In addition, traditional imaging technology, such as confocal microscopy, is expensive and requires skilled operators to conduct water sampling.

Hyperspectral imaging (HSI) has the advantage of simultaneously acquiring spectral information and two-dimensional spatial information [5]. Optical technology has been developed to be used widely in various biological areas and engineering applications, such as food quality and safety, medicine research and other fields [6]. HSI has the capability to rapidly detect and noninvasive fluorescence signatures of samples. It can also acquire a "hypercube", which is described as I(x, y, λ) and can be viewed either as a separated spatial image I(x, y) at each wavelength or as a spectrum I(λ) at each pixel. HSI has four scanning methods: point scanning, line scanning, area scanning and single shot scanning [7]. Compared with point scanning, line scanning operates precisely and concisely and also has an exposure time that is approximately 50 times shorter when scanning the same region for a constant excitation density [8]. The line scanning method records an entire line of image as well as the spectrum of each line of interest of the region. With the advantage of acquiring two-dimensional images across a range of spectra, HSI conjoining spectral features with spatial information are able to estimate specific features of samples [9]. Shao, Y et al. [10] verified the feasibility of detecting carotenoids in microalgae in situ using visible/near-infrared spectroscopy. By combining optical probe and chemometric methods, the result achieved processing discrimination power for certain algae. However, this method cannot capture the spatial information of microalgae. Davis et al. [11] found that hyperspectral florescence microscopy successfully detected and quantified the content of carotenoids and other photosynthetic pigments in microalgae [12]. Haaland et al. [13] achieved improved performance using multivariate curve resolution to extract quantitative information from hyperspectral images, resolved the overlapping spectrum and obtained pure components and concentrations of mouse macrophage cells. It decomposed into two pure components, autofluorescence and green fluorescent protein (GFP).

In this paper, a hyperspectral imaging microscope mainly based on microalgal-specific fluorescence signatures is used as a nondestructive detection system. The purpose is to explore a rapid and reliable method for varying spectral features in microalgae images and corresponding discovered species and to map the relative pigment content. The proposed method has the following advantages:

1. HSI microscopy is the combination of spectroscopy and digital imaging for spectral imaging of single cells containing many spectra;

2. HSI is optimized based on specific fluorescence signatures used in microalgae;

3. HSI is easy to implement, fast and reliable. HSI was combined with MCR to analyze pigments in microalgae.

## Materials and methods

### Algae strain and cultivation condition

Specimens and culture solutions were provided by Freshwater Alga Culture Collection at the Institute of Hydrobiology. The specimens encompassed Microcystis aeruginosa (FACHB-1338, Wuhan, China), Chlorella (FACHB-484, Wuhan, China), and Anabaena flosahuas (FACHB-245, Wuhan, China) for the experiment. BG11 was the culture solution that provides nutrition for cell growth. The microalgae were cultured in an incubator with the light and dark times set separately as 12 hours in one cycle. The temperature of the incubator was 25±2°C,

and the light intensity was 1000–2000 lux. The sample was divided into three groups according to species for measurement and imaging. In the experiment, a sample of 4–5 mm3 from the culture solution was placed in a glass slide, which was sealed by a cover slip with a thickness ranging from 0.13 mm to 0.17 mm. All experiments were carried out in a dark environment.

## The coordination between stage and camera

Hyperspectral imaging technology has the ability to collect the complete visible emission spectrum from microscope slides [5]. Synchronously controlling the movement between the motorized stage ((Westage, Shanghai, China) and charge-coupled device (CCD) is crucial to the imaging system. To acquire a full image focused on the region of interest, a slide is placed on the motored stage with a precise step by step movement value of 0.67 μm. The microscope stage and CCD (PRINCETON INSTRUMENT ISOPlane-160, the United States) are controlled with LabVIEW software (National Instrument, the United States). After a picture in the region of interest is acquired, the stage automatically moves to the next line in the region of interest and records the corresponding information in a second Y-λ file. Each Y-λ image is saved in a single file for each row along the excitation line of the sample [2]. With an exposure time of 100 milliseconds and 512 step movements, the data is obtained as a sequence of pictures and 512 files are saved. To show the spatial and spectral information simultaneously, the data is processed through spectral channel separation along wavelength and image fusion.

## MCR modeling algorithm

The MCR-ALS algorithm was used to extract various spectral profiles of pure components from unknown mixtures in measured samples. In the estimated data, the number of pure components was calculated by the MCR-ALS algorithm, which is based on the formula $D = CS^T + E$ (Eq 1) [14]. In the formula, the D matrix is experimental data, C is a concentration matrix, S is a spectrum matrix, and E is the matrix of residuals that is not explained by the resolved components. MCR-ALS can also be used in multiset analysis. Eq 1 is changed to the following formula:

$$\begin{bmatrix} D_1 \\ \vdots \\ D_n \end{bmatrix} = \begin{bmatrix} C_1 \\ \vdots \\ C_n \end{bmatrix} S^T + \begin{bmatrix} E_1 \\ \vdots \\ E_n \end{bmatrix} \tag{1}$$

In this case, a common pure spectrum matrix $S^T$ and several different matrices containing independent concentration profiles (S1, . . ., Sn) were resolved.

Prior to the MCR analysis, the image data (consisting of two dimensions and a spectral dimension presented in this paper) were unfolded into a two-dimensional matrix at the individual spectra. MCR was implemented using a non-negativity constraint in which outputs were pure-emission spectra of the independently varying components and relative concentration maps. After the analysis, MCR concentrations from each pure-spectral component were reshaped into the spatial domain to create concentration images for each component. The initialization process consisted of:

1. Define the number of PCA factors to use in the noise, data size and reduction step prior to MCR analysis.

2. Set the number of MCR interactions performed.

3. Determine the number of spectral components presented in the data set.

4. Provide the MCR algorithm with the initial spectral estimates.

Once the number of spectral components was determined, MCR analysis was initiated with the appropriate number of pure spectra. Each value consisted of a uniform random number between zero and one, which was normalized to unit length during the data processing.

In the optimization process, the C and ST matrices need to be estimated initially. Calculating the values in two cases: (1) when C is a constant, find $S^T$ at the minimum value of formula 2, (2) when $S^T$ is a constant, find C at the minimum value of Eq 2:

$$|I^{PCA} - CS^T| \tag{2}$$

$I^{PCA}$ is a data matrix composed of principal component analysis (PCA) using a specified number of principal components [15].

In conclusion, the method consisted of three steps:

1. Calculate the covariance of the original number under the PCA, the eigenvalue and the characteristic matrix. Furthermore, the appropriate number of principal components was selected according to the contribution rate of the principal component;

2. Calculate the eigenvalue of the forward matrix and the reverse matrix. The positive and reverse eigenvalues were taken as asymptotic factors to obtain the concentration function and determine the start and end positions.

3. The eigenvalues obtained in the previous step were plotted as the initial component matrix estimate. The measurement matrix was repeatedly iterated using this matrix. After setting the number of iterations, the pure spectrum and component concentration were obtained.

To evaluate the performance of the model, a reference value lacked fit and the correlation coefficient of prediction. The equation was defined as follows:

$$\text{Error of data fit}\,(\%) = 100 \sqrt{\frac{\sum_{ij} e_{ij}^2}{\sum_{ij} d_{ij}^2}} \tag{3}$$

where $d_{ij}$ is an element of the input data matrix D and $e_{ij}$ is the related residual obtained from the difference between the input element and the MCR-ALS reproduction.

The optimum number of components was obtained by checking the performance of fitting for the MCR model [16]. If the estimated data contain negative values, the purest variable selection algorithm does not work. A nonnegative least squares (NNLS) algorithm was used that includes row constraints and column constraints, which limits the data in the appropriate region.

## Relationship between fluorescence intensity and pigment content

The original image is a three-dimensional data cube that can be expressed as F (x, y, λ), where x and y represent spatial dimension information, and λ represents spectral dimension information. The pixel value f(x, y, λ) of the region of interest in each band was determined by the LED light source spectrum S(λ, t), the transmittance function H(x, y, λ, t) of the specific position of the sample, the transmission characteristic L (λ, t) of the filter and the response

function M $(x, y, \lambda, t)$ of the CCD. According to the following equation [19]:

$$f(x, y, \lambda) = \int_0^t S(\lambda, t)H(x, y, \lambda, t)L(\lambda, t)M(x, y, \lambda, t) \tag{4}$$

In the formula, t represents the time to collect an image, which was mainly composed of the exposure time and readout time of the CCD. Since a short amount of time was required to collect one image, the time was far less than the total time for collecting images and was ignored in the data. According to the Beer-Lambert law, incident light of a certain wavelength can be used to determine the relationship between the solution absorbance and the concentration c and optical path b:

$$I_a = p \cdot c \cdot b \tag{5}$$

$$-\lg\varepsilon = I_a \tag{6}$$

where p is the absorbance coefficient. The unit of absorbance is a percentage when expressed as transmittance $\varepsilon$.

The pigment in the microalgae was excited after absorbing light energy and emitted fluorescence. The fluorescence intensity of the algae fluid is related to which fluorescent substance in the algae fluid absorbs the light energy and the fluorescent efficiency. In other words, the fluorescent pigment in the algae was excited by the incident light $I_0$ to produce fluorescence F. The concentration of the fluorescent substance in the algae fluid was c, and the thickness of the liquid layer was l. Since the fluorescence intensity F is proportional to the light intensity absorbed by the fluorescent substance, it can be expressed by a linear equation:

$$F = \varphi_f I_a \tag{7}$$

Therefore, the formula was concluded by the equation:

$$I_a = I_o(1 - 10^{Ec1}) \tag{8}$$

With respect to Ecl< 0.05, the equation was approximately equal to:

$$F = \varphi_f I_o(1 - e^{-2.303Ec1}) \tag{9}$$

$$F = \varphi_f I_o(1 - e^{-2.303Ecl}) \tag{10}$$

The value of K was determined by the fluorescence efficiency of the fluorescence process under certain conditions. For a very low algae concentration, the fluorescence intensity was linearly related to the concentration of the fluorescent substance.

## Results

### Hyperspectral fluorescence imaging system based on line scanning

The hyperspectral **fluorescence** imaging system has hardware and software. The hardware consisted of an epifluorescence microscope (LYMPUS-IX73P2F, Japan), electric-motorized stage (Westage, Shanghai, China), imaging spectroscopy (PRINCETON INSTRUMENT ISO-Plane-160, the United States) and data acquisition (NI USB-6361, the United States). In the result, the line-scanning was mainly realized by the movement of electric-motorized stage. The microscope has two light sources (Fig 1), a halogen lamp used for transmitted light, and a light emitting diode used for excitation. A broad-pass excitation filter (Olympus, U-MWU, Japan) was used to restrict background fluorescence. The objective was 100X with an NA of 1.40

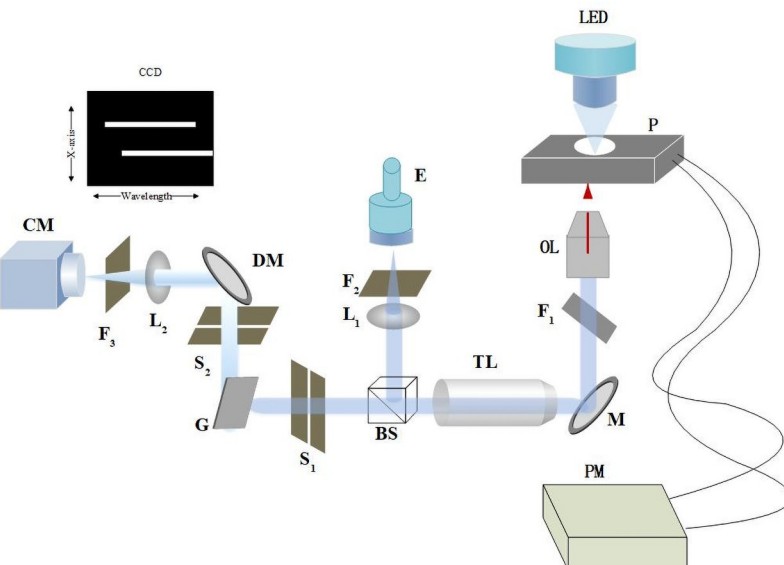

**Fig 1. Optical path diagram.** From the light source (LED), the light line passes the fast filter wheel (F1) and lens (L1) and reaches the samples. The light focuses on and illuminates the samples through an objective (OL). PM is the pulse modulator, which controls the motored stage step by step. The light beam is adjusted through a tube lens (TL) and beam splitter (BS). A line of the light illuminates the sample region in the same position as the resulting emission light through a diorchic mirror (DM) and the imaging spectroscope, and finally reaches the CCD camera (CM).

(Olympus UPLSAPO100XO, Japan) and 40X with an NA of 0.75 (Olympus UPLFLN 40X). Imaging spectroscopy is used to split light into a serial spectrum in which the wavelength ranges from 350 nm to 800 nm, which consists of spectroscopy and imaging sensors. Spectroscopy contains two gratings with a blaze wavelength of 600 nm and grooves of 300 line pairs per mini-meter (lpmm) and 1200 lpmm. The CCD camera has an image resolution of 1024*1024 pixels, with one pixel being 13 μm*13 μm. The motorized stage has a repositioning accuracy of 50 nm in theory and 51 nm in the measured experiment in S1 Table. An entrance slit of 80 μm was set under illumination with a light source, which corresponds to 0.84 times the diameter of the Airy disk at 680 nm [9]. Therefore, when the image is acquired along the X axis from the CCD camera without binning, it corresponds to a sampling of fluorescent intensity at a single step of 0.67 μm under 100X Objective.

## Hyperspectral images of the microalgal pigment

An image is captured according to the system discussed above (raw data). There are two image preprocessing steps in Fig 2. The first step is to calculate the total intensity image based on the integral method. The average image is divided by the total intensity image. Moreover, by searching every pixel of the image, the position of spectral information that is saved in a matrix can be determined. The precolor image is colored by this matrix.

The precolor image shows in Fig 3. Each color in the image represents different wavelengths. Anabaena flosahuas in Fig 3(a) shows its heavy fluorescence intensity of pigment protein. The fluorescence intensity has sharp edges in the cell. It is obvious that the intensity distribution of pigment is homogeneous and mainly distributed along the filamentous direction. The green region is 685 nm, corresponding to phycobilisome (PBS), and the yellow region is 700 nm, corresponding to chlorophyll, as shown in Fig 3(a). For Microcystis

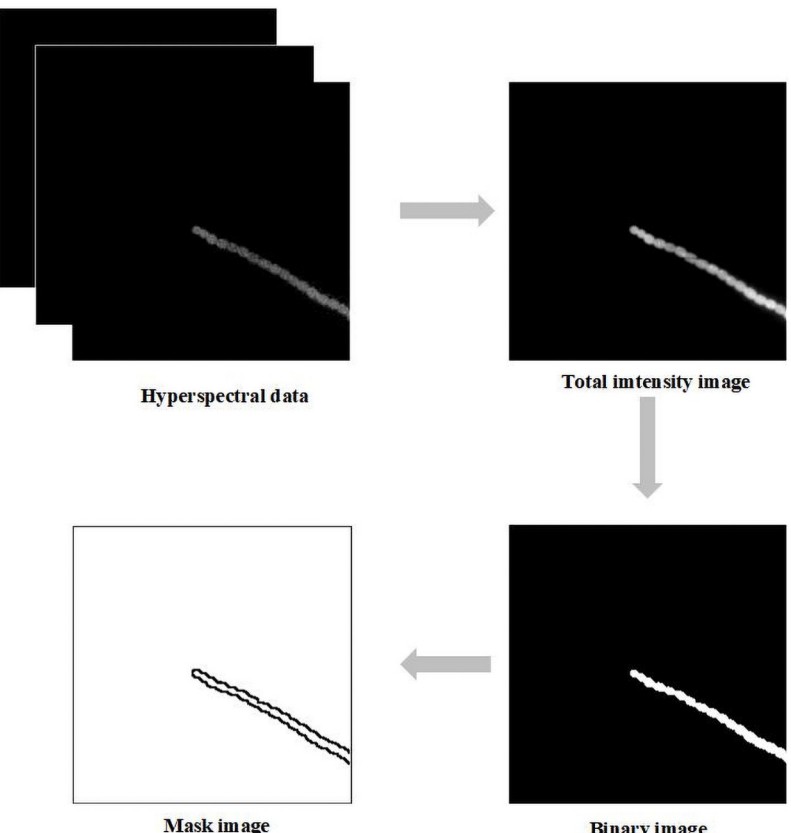

**Fig 2. Image preprocessing.** After image reconstruction, an integral method was used to accumulate the total intensity in each image of the data file. Binary values are used to replace the pixel values of the region of interest with one and replace the pixel values of background with zero. Finding edges based on binary images generates a mask image.

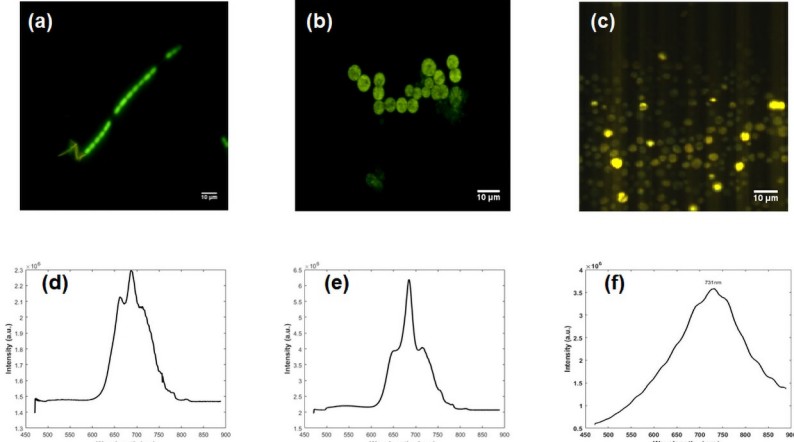

**Fig 3. Hyperspectral fluorescence images and extraction spectrum.** (a) Precolor overlay image of Anabaena flosahuas. (b) Precolor image of Microcytic aeruginosa (c) Color overlay image of Chlorella sp. (d) Mean spectrum of Anabaena flosahuas (e) Mean spectrum of Microcytic aeruginosa. (f) Mean spectrum of Chlorella sp.

aeruginosa, the peak at 660 nm comes from chlorophyll of photosynthesis system II (PSII), and a subpeak over 710 nm is produced in chlorophyll among photosynthesis system I (PSI).

The second image preprocessing step is to convert the average image into a grayscale image using the function "rgb2gray" in MATLAB in the Fig 2 [12]. The binary image is operated on a gray image by a threshold of a value of 0.5. The mask image is determined using the edge function by removing the background instead of white pixels. The mask area has a higher signal than other regions. This processing is important for further analysis to extract useful spectra in Fig 2. This is because the accuracy of pigment analysis is relative to the ratio of signal intensity to noise [17,18]. In fact, spectra are very similar when they are obtained from different pixels within the region of interest for the same species.

In addition, the spectrum of Chlorella sp. has a peak at a wavelength of 470–574 nm in the emission spectrum in Fig 7(f). However, carotenoids do not appear in Fig 3(f). This is because the intensity of chlorophyll at the peak of 680 nm was greater than that of carotenoids at the average intensity ranging from 470 nm to 574 nm. Carotenoids were measured to only encompass 1/200 of the intensity at 680 nm. When the spectrum has a high degree of overlap, the fusion image method is not available for pigment analysis.

The average spectra of Microcystis aeruginosa and Anabaena flosahuas are similar in shape, but they are different in the wavelength bands from 650 nm to 710 nm. Microcystis aeruginosa has a single peak at 683 nm. Anabaena flosahuas has dual fluorescence peaks at 660 nm and 685 nm, with a shoulder peak at 710 nm. In fact, the type of substance cannot be inferred and further analysis is necessary.

## Reconstruction image based on the MCR method

To reveal the spectral features and location of fluorescent pigments, MCR analysis is used to computationally isolate the pure spectral components that contribute to the fluorescence data and develop a spectral model describing the results.

The independent image based on the MCR model in Fig 4 shows the concentration distribution and corresponding pure spectra of cyanobacteria. The color bar represents the relative concentration, and the color scale is consistent in all of the main images to permit direct comparison of the colors and intensities to assess the relative concentration. Red indicates a high concentration, and blue indicates a low concentration. For Anabaena flosahuas, two main pigments assigned to PBS and Chl-a were resolved from the MCR model. The fluorescence

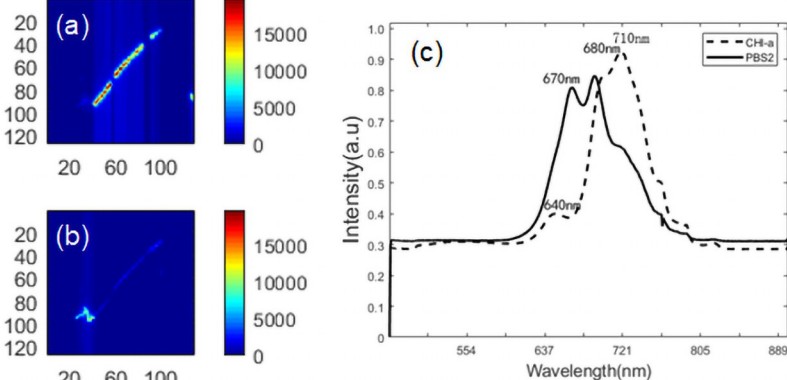

**Fig 4. MCR analysis of algal cells for Anabaena flosahuas.** (a) Concentration image of PBS resulting from MCR analysis of the entire spectral area. (b) Concentration distribution images of Chl-a resulting from analysis of the entire spectral area using MCR. (c) Pure spectral profile of chlorophyll (dotted line) and PBS (solid line).

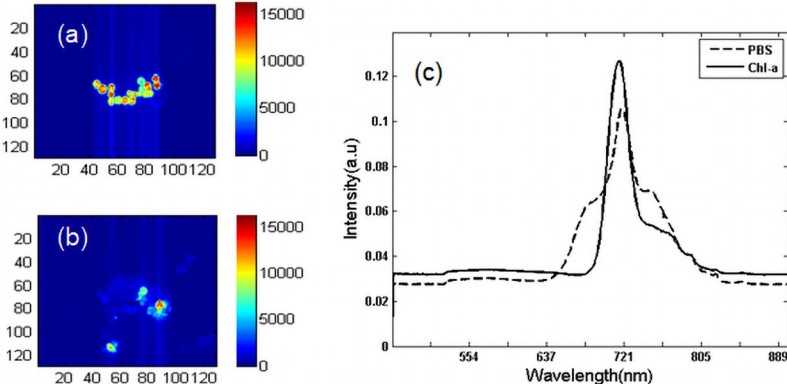

**Fig 5. MCR analysis of Microcystis aeruginosa.** (a) Concentration image of PBS resulting from analysis of the entire spectral area using MCR. (b) Concentration image of CHl-a resulting from analysis of the entire spectral area using MCR. (c) MCR pure-component spectrum containing the PBS component (dotted line) and chlorophyll-a (solid line). Color scales are identical for all images to facilitate comparison between images. The x-axis and y-axis of the image represent image size (pixels).

distribution of Anabaena flosahuas shows that the fluorescence is less strong at the edge of the cell than at the inner center of the cell (Fig 4). The peaks of the resolved spectra for Anabaena flosahuas corresponding to 640 nm, 710 nm, 670 nm and 680 nm are attributed to phycocyanin (PC), PSI, allophycocyanin (APC) and PSII, respectively, in Fig 4(c). An interesting location pattern is observed. The Chl-a was uniformly distributed in all cells (Fig 4(b)), while the pigment protein was concentrated at the cell junctions (Fig 4(a)) and only discernible in a few cells.

For Microcystis aeruginosa, the fluorescence emission spectrum of the resolved spectra observed at 721 nm is similar in Fig 5(c). However, the spectrum between the resolved spectrum and unresolved spectrum is different from the peak. It should be noted that the spectrum shows a redshift action ranging from 683 nm (in Fig 3(c)) to 721 nm in Microcystis aeruginosa. The main two components in Microcystis aeruginosais are PBS and Chl-a, which are represented by the dotted lines and solid lines in Fig 5(c) through their shape and intensity. The images corresponding to the two photosynthetic pigments were pseudocolored and overlaid for visualization purposes.

## Optimization of pigment analysis

We know that the green algae comprises several different photosynthetic pigments, such as carotenoids and chlorophyll, that are distributed in a specific region of the cell. For Chlorella sp., the two resolved spectra are not consistent with the three images obtained by the MCR model in Fig 6(c). There are two factors that can be attributed to this problem. First, the fluorescence intensity of chlorophyll is always much greater than that of carotenoids in cells [19]. In addition, the 685 nm emission peak is the dominant spectral feature in Fig 3(e). We verified the emission measurements using the methanol extraction method. The emission spectrum in detail specifically represents the existence of carotenoids ranging from 470 nm to 574 nm.

The special optimization process is as follows:

To address the problem of missing carotenoids, we restricted the wavelength of MCR analysis in the carotenoid spectral region (470–574 nm) joining the spectrum selection method (Fig 7). As a result, we were able to obtain the carotenoid component spectrum and the relative concentration image. In the spectrum selection, the carotenoid spectrum can be extracted by

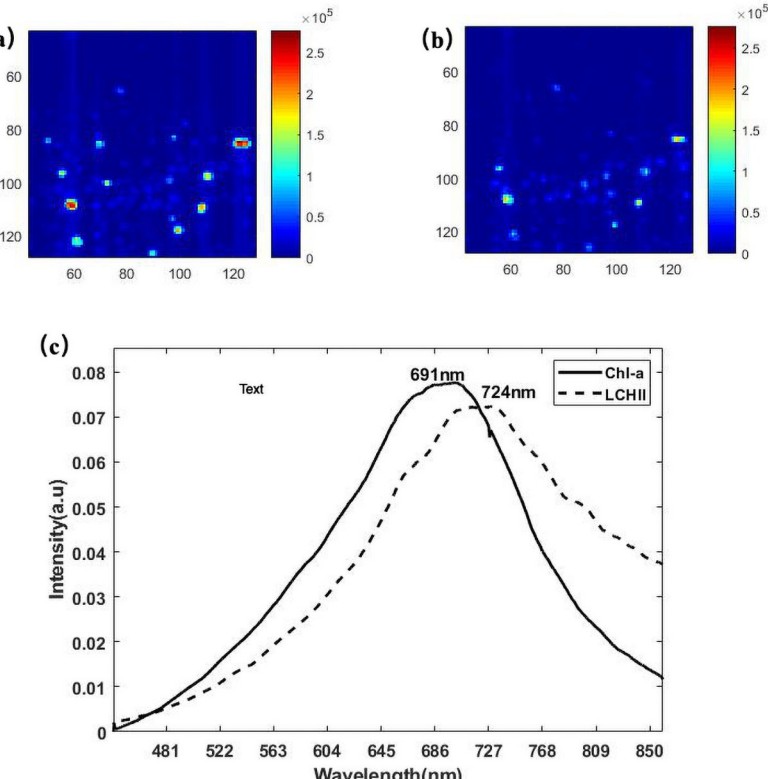

**Fig 6. MCR analysis of Chlorella sp. for the entire spectrum.** (a) Concentration image of carotenoids resulting from analysis of the entire spectral area using MCR. (b) Concentration image of CHl-a resulting from analysis of the entire spectral area using MCR. (c) MCR pure-component spectrum containing the PBS component (solid line) and chlorophyll-a (dotted line). Color scales are identical for all images to facilitate comparison between images. The x-axis and y-axis of the image represent image size (pixels).

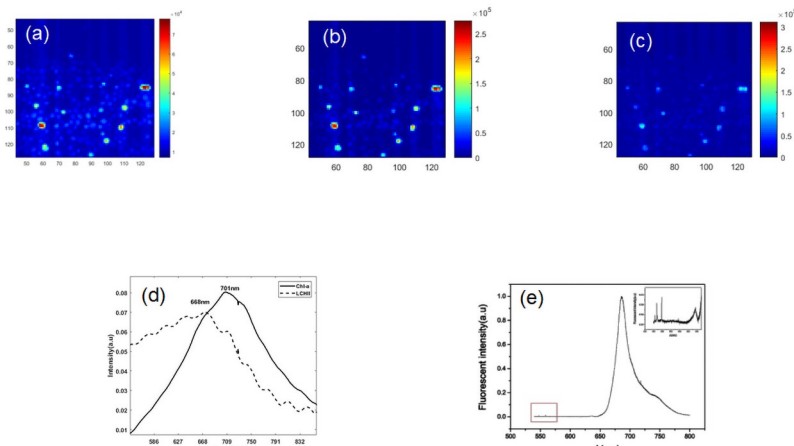

**Fig 7. MCR analysis of Chlorella sp. for spectrum selection.** (a) Concentration images for Chlorella sp. of carotenoid resulting from analysis of selective spectral area (470–574 nm) (b) Concentration images of Chlorophyll a (c) Concentration images of Light-Harvesting Complex II(LCH-II). (f) pure component spectrum of Chlorella sp. representing chlorophyll a (solid line) and light-harvesting complex II (dotted line) (e) Emission measurement under pigment extraction above 574 nm excited at 512 nm (ex: 512 nm). This describes the intensity difference between carotenoid and chlorophyll spectral areas.

restricting wavelengths greater than 550 nm to zero. The relative concentration image of carotenoids was consistent with the image and was calculated by spectrum selection in detail. The image describes the carotenoids mainly distributed in the internal cell membrane of the algae.

## Discussion

Microalgae contain many kinds of pigments and proteins in the membrane, where light reactions occur among these fluorophores in the center of the photosynthetic system. Additionally, an energy transfer mechanism occurs in the region distributed from the thylakoid membrane to chloroplasts. Electron transfer takes place in the reaction center when a chlorophyll molecule transfers an electron to a neighboring pigment molecule. Pigments and proteins involved in this primary electron transfer define the reaction center [20,21].

Hyperspectral imaging is an imaging spectroscopy technique that combines imaging technology with spectroscopy to detect biological samples. A spectral image contains continuous spectral information, one image for each individual pixel of the sample. Each image is acquired by dividing the spectral image into a set of single wavelengths or a narrow range of wavelengths. The wavelengths at different channels can also be fused into multidimensional images to indicate the spectral distribution in an image. Additionally, the content of constitution in the sample can be estimated based on the intensity of the spectrum.

### Verification experiment

As a verification experiment, there are emission measurement and absorption measurement. The emission spectrum was detected by a fluorescence spectrophotometer (SHIMADAZI 5301). *In vivo*, the fluorescent excitation spectrum of Chlorella sp. cells indicates that the pigment composition is Chl-a (671 nm) and carotenoid (492 nm). This result (Fig 8(a)) is consistent with the emission spectrum (Fig 7(d)) extracted from hyperspectral data. However the emission measurement cannot detect directly the pigment of peak at 680nm.

For Anabaena flosahuas, the fluorescence excitation spectrum was similar to that of most other cyanobacteria. With a large amount of phycobiliproteins for cyanobacteria, spectral peaks corresponding to 591 nm and 710 nm were obtained. The absorption measurement using a spectrometer (PerkinElmer Lambda950) was performed at the same time and under the same conditions as the HSI measurement. The data shows in the S1 Fig. The wavelength ranged from 350 nm to 800 nm. The absolute concentrations of pigments such as chlorophyll (Chl) and phycobilin (PC) and allophycocyanin (APC) were calculated using the following

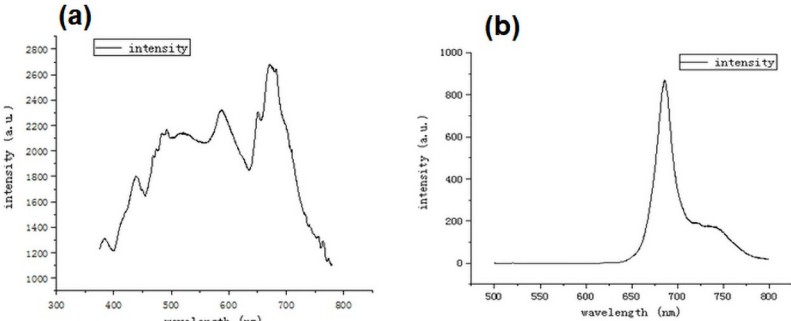

**Fig 8.** (a) Emission spectrum and (b) absorbance spectrum of Chlorella sp. Emission measurement under pigment extraction excited at 512 nm (ex: 512 nm). The peak is at 671 nm, 587 nm, 492 nm and 438 nm. This describes the intensity difference between carotenoid and chlorophyll spectral areas.

equations [22]:

$$Chl = 14.96(A678 - A730) - 0.616(A625 - A730) \tag{11}$$

$$PC = (A620 - 0.71A650)/6.83 \tag{12}$$

$$APC = (A650 - 0.19A620)/5.99 \tag{13}$$

For verification, the calculated concentration shows in S2 Table In conclusion, it is accordance with the data reconstructed by the method of MCR-ALS.

## Conclusion

With a high degree of overlapping spectra, the distribution of pigment in the region of interest remains unknown according to a previous report. The method not only overcomes the limitation of conventional technology to detect unknown pigments, but also it allow for acquiring the relative concentration image and resolving the overlapping spectra into pure components of single pigment [20,23]. For this work, hyperspectral imaging combined with the MCR algorithm simultaneously got the spectral features and pigment location. The spectral features of the pigment components are consistent with the fluorescence excitation spectrum in the verification experiment. In addition, when pigment analysis misses a component, spectrum selection plays an important role and allows for varying fluorescent components that account for low intensity. For Anabaena flosahuas, Chlorella sp. and Microcystis aeruginosais which cause water pollution, the lack of fit of prediction was 1.9151, 1.4875 and 0.3942, respectively. The visualization performance for Chlorella sp. and Anabaena was better than that for Microcystis aeruginosa. The concentration of adherent cells or algae with a low level of autofluorescence intensity is not estimated. The optimization of imaging systems will lead to more in-depth research in the future.

## Supporting information

**S1 Fig. Absorption spectrum of 147 samples.**
(PDF)

**S2 Fig. Chlorella sp. of N-normal condition.** (a)fusion image (b)pure component spectrum.
(PDF)

**S3 Fig. The images of cyanobacterial algae under different concentration.** (a) overlapped and concentrated cells (b) sparse cells.
(PDF)

**S1 Table. System parameter.**
(PDF)

**S2 Table. The concentration of pigment.**
(PDF)

## Author Contributions

**Conceptualization:** Lijin Lian, Xuejuan Hu.

**Data curation:** Lijin Lian.

**Formal analysis:** Lijin Lian, Xuejuan Hu, Zhenhong Huang.

**Funding acquisition:** Xuejuan Hu.

**Investigation:** Zhenhong Huang, Liang Hu, Lu Xu.

**Methodology:** Lijin Lian, Zhenhong Huang, Lu Xu.

**Project administration:** Xuejuan Hu.

**Software:** Lijin Lian, Zhenhong Huang.

**Supervision:** Xuejuan Hu.

**Validation:** Lijin Lian, Zhenhong Huang.

**Visualization:** Lijin Lian, Liang Hu.

**Writing – original draft:** Lijin Lian.

**Writing – review & editing:** Xuejuan Hu, Zhenhong Huang, Liang Hu, Lu Xu.

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
