## [Decision Letter · Decision Letter 0]

22 May 2021

PONE-D-21-07134

Pigment analysis based on a line-scanning fluorescence hyperspectral imaging microscope combined with multivariate curve resolution

PLOS ONE

Dear Dr. Hu,

Thank you for submitting your manuscript to PLOS ONE. After careful consideration, we feel that it has merit but does not fully meet PLOS ONE’s publication criteria as it currently stands. Therefore, we invite you to submit a revised version of the manuscript that addresses the points raised during the review process.

We look forward to receiving your revised manuscript.

Kind regards,

Xiaowei Zhang, Ph.D.

Academic Editor

PLOS ONE

Journal Requirements:

PLOS requires an ORCID iD for the corresponding author in Editorial Manager on papers submitted after December 6th, 2016. Please ensure that you have an ORCID iD and that it is validated in Editorial Manager. To do this, go to ‘Update my Information’ (in the upper left-hand corner of the main menu), and click on the Fetch/Validate link next to the ORCID field. This will take you to the ORCID site and allow you to create a new iD or authenticate a pre-existing iD in Editorial Manager. Please see the following video for instructions on linking an ORCID iD to your Editorial Manager account: https://www.youtube.com/watch?v=_xcclfuvtxQ

We thank XueJuan Hu Lab members for sharing instruments and discussion and the Institute of Hydrobiology provided the microalgal cells. The research was funded by Natural Science Foundation of Guangdong Province (2017A030310568) and Science, Technology and Innovation Commission of Shenzhen Municipality (JCYJ20180301170959233)

the funders had no role in study design, data collection and analysis, decision to publish, or preparation of the manuscript.

5. We note you have included a table to which you do not refer in the text of your manuscript. Please ensure that you refer to Table 1 and 2 in your text; if accepted, production will need this reference to link the reader to the Table.

Reviewers' comments:

Reviewer's Responses to Questions

**Comments to the Author**

1. Is the manuscript technically sound, and do the data support the conclusions?

Reviewer #1: Yes

Reviewer #2: Yes

2. Has the statistical analysis been performed appropriately and rigorously? 

Reviewer #1: Yes

Reviewer #2: Yes

3. Have the authors made all data underlying the findings in their manuscript fully available?

Reviewer #1: Yes

Reviewer #2: Yes

4. Is the manuscript presented in an intelligible fashion and written in standard English?

Reviewer #1: Yes

Reviewer #2: Yes

5. Review Comments to the Author

Reviewer #1: In this manuscript, the authors aim to algae detection that use hyperspectral fluorescence imaging to capture images and detect specific fluorescence signatures. Overall, the study provides a new approach to detect the algal pigment. This manuscript can be considered for publication after proper revisions.

1. “Another approach is to capture images based on specific fluorescence signatures [3]. That means that the value of fluorescent peaks is considered as the gold standard for pigment analysis” (lines 33 and 34). What does the gold standard mean? and is it another approach HSI?

2 The author argued that ‘HSI is easy to implement, fast and reliable (line 68). Why?

3. Sometimes, the authors use the abbreviation without definition.

4. In line 189 ‘Spectroscopy contains two gratings with a blaze wavelength of 600 nm and grooves of 300 mm-1 and 1200 mm-1’, the unit of grating is not correct.

Reviewer #2: In this manuscript, the authors carry on the analysis in the algae based on fluorescent hyperspectral imaging microscope. The results realize the visualization of the microalgae pigment obtained from reconstruction image and demonstrates the feasibility of applying the method to analyze the algae. The results are interesting and reliable. In my opinion, the manuscript can be accepted for publication in PLOS ONE after some minor points are addressed.

1. the line 176 ‘For a very low algae concentration, the fluorescence intensity was linearly related to the concentration of the fluorescent substance’ is the basic principle to calculate the content of pigments. However, the value of low algae concentration could not be found from the whole manuscript, and it should be given in revised manuscript.

2. The description and presentation of results aren’t clear. For example, ‘For this work, hyperspectral imaging combined with the MCR algorithm simultaneously got the spectral features and pigment location. The spectral features of the pigment components are consistent with the fluorescence excitation spectrum in the verification experiment.’ It is also not clear for the data of verification experiment. Please make description in detail. (lines 338 and 339).

3. Some expressions should be refined by native English speakers.

6. PLOS authors have the option to publish the peer review history of their article (what does this mean?). If published, this will include your full peer review and any attached files.

Reviewer #1: No

Reviewer #2: No

---

## [Author Response · Author response to Decision Letter 0]

17 Jun 2021

Response to Reviewers1:

Reviewer #1: 

1. “Another approach is to capture images based on specific fluorescence signatures [3]. That means that the value of fluorescent peaks is considered as the gold standard for pigment analysis” (lines 33 and 34). What does the gold standard mean? and is it another approach HSI?

Answer Q1:

Another approach in this sentence(lines 33 and 34) is exactly the specialized microscope. The manuscript presents the hyperspectral imaging microscope based on the line-scanning method. The method acquires the spatial and spectral information and reconstructs the image according to the fluorescent feature. Therefore another approach in the paper is based on the fluorescent feature to catch the images of samples. 

2 The author argued that ‘HSI is easy to implement, fast and reliable (line 68). Why?

Answer Q2:

Hyperspectral imaging microscope is a method with the advantages of acquiring spatial and spectral information simultaneously. The system contains two parts: the hardware and the software. We programmed the user interface to control the movement of stage and capture of images. It is concise and easily operated. As soon as just pressing the button, we acquire the hyperspectral images of samples. Moreover, the system is based on the line scanning, that is, scanning the sample each time we can get spectral information of a whole line. It costs time shorter than the point-scanning method.

3. Sometimes, the authors use the abbreviation without definition.

Answer Q3:

From the whole manuscript, I found the lines 60 and 99 has a abbreviation without definition. The GFP is the abbreviation of green fluorescent protein; the MCR-ALS algorithm is the abbreviation of multivariate curve resolution - alternating least squires. The MCR-ALS algorithm develops based on the MCR algorithm.

4. In line 189 ‘Spectroscopy contains two gratings with a blaze wavelength of 600 nm and grooves of 300 mm-1 and 1200 mm-1’, the unit of grating is not correct.

Answer Q4:

The unit of grating isn't mm-1 but lpmm, which is the abbreviation of line pairs per mini-meter. It means that the grating has two gratings. One is 300 line pairs per mini-meter in the surface. Anther is 1200 line pairs per mini-meter. 

Response to Reviewers2:

Reviewer #2: 

1.the line 176 ‘For a very low algae concentration, the fluorescence intensity was linearly related to the concentration of the fluorescent substance’ is the basic principle to calculate the content of pigments. However, the value of low algae concentration could not be found from the whole manuscript, and it should be given in revised manuscript.

Answer Q1: 

Before the experiment, we diluted the samples with the original solution to culture medium at a ratio of 1: 1000 in the sample preparation. The sample has more than the original concentration of 10e6 cells per milliliter. Moreover, in order to verify the limited numbers of microalgal cells we can catch the images of algae under different concentration. It concludes that the detection limited is more than two cells has enough intensity that could be detected. The image in S3 figure is full of cells, which isn’t classified for several cells. What‘s more, we catch the image without overlapping cells that can be distinguished. 

2. The description and presentation of results aren’t clear. For example, ‘For this work, hyperspectral imaging combined with the MCR algorithm simultaneously got the spectral features and pigment location. The spectral features of the pigment components are consistent with the fluorescence excitation spectrum in the verification experiment.’ It is also not clear for the data of verification experiment. Please make description in detail. (lines 338 and 339).

Answer to Q2:

In the verification experiment, we have got the data of the absorption spectrum and the emission spectrum using 95% alcohol to extracting the pigment in algae. Emission measurement excited at 512 nm (ex: 512 nm) after pigment extraction. The peaks of emission spectrum are at 671 nm, 587 nm, 492 nm and 438 nm(figure 8), which is consistent with the peaks of 675nm and sub-peak ranges from 470nm to 574nm respectively in the hyperspectral images(Fig S2). The fluorescent excitation spectrum of Chlorella sp. indicates that the pigment composition is Chl-a (671 nm) and carotenoid (492 nm). Therefore，the spectral feature of the pigment components are consistent with the fluorescence excitation spectrum.

3. Some expressions should be refined by native English speakers.

Answer to Q3:

The manuscript is revised through native speaker in spelling, English-grammar and language usage. I employed the professional scientific editing service to check the the text.

Thanks for your good comments.

---

## [Editor Report · Decision Letter 1]

6 Jul 2021

Pigment analysis based on a line-scanning fluorescence hyperspectral imaging microscope combined with multivariate curve resolution

PONE-D-21-07134R1

Dear Dr. Hu,

We’re pleased to inform you that your manuscript has been judged scientifically suitable for publication and will be formally accepted for publication once it meets all outstanding technical requirements.

Kind regards,

Xiaowei Zhang, Ph.D.

Academic Editor

PLOS ONE

Additional Editor Comments (optional):

The author's responses to the reviewer's queries are satisfactory. Now the revised version improves well with the better scientific discussions. I consider the manuscript is suitable for publication
---

## [Editor Report · Acceptance letter]

30 Jul 2021

PONE-D-21-07134R1 

Pigment analysis based on a line-scanning fluorescence hyperspectral imaging microscope combined with multivariate curve resolution 

Dear Dr. Hu:

I'm pleased to inform you that your manuscript has been deemed suitable for publication in PLOS ONE. Congratulations! Your manuscript is now with our production department. 

Kind regards, 

on behalf of

Dr. Xiaowei Zhang 

Academic Editor

PLOS ONE